# Long-Chain Acyl Coenzyme A Dehydrogenase, a Key Player in Metabolic Rewiring/Invasiveness in Experimental Tumors and Human Mesothelioma Cell Lines

**DOI:** 10.3390/cancers15113044

**Published:** 2023-06-03

**Authors:** Daniel L. Pouliquen, Giacomo Ortone, Letizia Rumiano, Alice Boissard, Cécile Henry, Stéphanie Blandin, Catherine Guette, Chiara Riganti, Joanna Kopecka

**Affiliations:** 1Université d’Angers, Inserm, CNRS, Nantes Université, CRCI^2^NA, F-49000 Angers, France; 2Department of Oncology, University of Torino, via Santena 5/bis, 10126 Torino, Italy; giacomo.ortone@edu.unito.it (G.O.); letizia.rumiano@edu.unito.it (L.R.); chiara.riganti@unito.it (C.R.); joanna.kopecka@unito.it (J.K.); 3Université d’Angers, ICO, Inserm, CNRS, Nantes Université, CRCI^2^NA, F-49000 Angers, France; alice.boissard@ico.unicancer.fr (A.B.); cecile.henry@ico.unicancer.fr (C.H.); catherine.guette@ico.unicancer.fr (C.G.); 4CHU Nantes, CNRS, Inserm, BioCore, US16, SFR Bonamy, Nantes Université, F-44000 Nantes, France; stephanie.blandin@univ-nantes.fr

**Keywords:** malignant mesothelioma, metabolism, mitochondria, long-chain specific acyl-CoA dehydrogenase, fatty acid β-oxidation, biomarker

## Abstract

**Simple Summary:**

This study aims to investigate mitochondrial metabolic differences between invasive and non-invasive malignant mesotheliomas in order to find new biomarkers for invasive properties and new potential actionable targets with the goal of improving the diagnosis and treatment of such tumors, which are highly resistant to current treatments.

**Abstract:**

Cross-species investigations of cancer invasiveness are a new approach that has already identified new biomarkers which are potentially useful for improving tumor diagnosis and prognosis in clinical medicine and veterinary science. In this study, we combined proteomic analysis of four experimental rat malignant mesothelioma (MM) tumors with analysis of ten patient-derived cell lines to identify common features associated with mitochondrial proteome rewiring. A comparison of significant abundance changes between invasive and non-invasive rat tumors gave a list of 433 proteins, including 26 proteins reported to be exclusively located in mitochondria. Next, we analyzed the differential expression of genes encoding the mitochondrial proteins of interest in five primary epithelioid and five primary sarcomatoid human MM cell lines; the most impressive increase was observed in the expression of the long-chain acyl coenzyme A dehydrogenase (ACADL). To evaluate the role of this enzyme in migration/invasiveness, two epithelioid and two sarcomatoid human MM cell lines derived from patients with the highest and lowest overall survival were studied. Interestingly, sarcomatoid vs. epithelioid cell lines were characterized by higher migration and fatty oxidation rates, in agreement with ACADL findings. These results suggest that evaluating mitochondrial proteins in MM specimens might identify tumors with higher invasiveness. Data are available via ProteomeXchange with the dataset identifier PXD042942.

## 1. Introduction

The role of mitochondria, at the crossroads of many studies related to cancer invasiveness, has been extensively investigated over the last fifteen years [1]. Their involvement in motility and invasion, microenvironment, plasticity, and colonization was recently reviewed [2]. Since the pioneering work of Ishikawa et al. demonstrating the role of mtDNA transfer in the acquisition of high metastatic potential [3], cancer cells were shown to acquire mitochondria from neighboring cells in order to acquire phenotypic characteristics, including stemness, representing a gain of function for tumors, i.e., enhancing their invasive properties [4]. The interplay between mitochondrial dynamics and extracellular matrix (ECM) remodeling was emphasized [5], and the molecular mechanisms linking dysregulated fission/fusion to tumor progression and metastasis were deciphered [6]. Together with an updated view of the effects of mitochondria dysfunction on tumor glycolysis [7], these important breakthroughs are having a profound impact on new therapeutic strategies aiming at overcoming hypoxic and chemorefractory tumors [8]. Finally, the upregulation of mitochondrial proteins involved both in ATP production and drug resistance [9], and/or immune-resistance [10] could lead to new therapies. In parallel, all these studies could also shed light on new biomarkers for better predictions of cancer chemosensitivity [11].

In this context, proteomics-based investigations provide crucial insights into the role of mitochondrial proteome rewiring [11,12,13]. The development of high-throughput proteomics techniques, combined with the use of experimental models of increasing invasiveness, has led to the identification of new proteins of interest both in rats and humans [14]. Given this methodological background, in this study, we aim to focus on mitochondrial proteins, which appear to play an important role in metabolic rewiring and the invasiveness process, both in experimental tumor models and tumor cells from patients.

## 2. Materials and Methods

### 2.1. Collection of Rat Tumor Tissues for Proteomic Analyses

The formalin-fixed paraffin embedded (FFPE) tissue samples used in this study were collected from the same groups of Fisher F344 rats with four different experimental mesotheliomas at increasing stages of invasiveness, as previously described [15]. To generate the tumors, the experimental procedures used for in vivo manipulations at the Unité Thérapeutique Expérimentale de l’Institut de Recherche en Santé de l’Université de Nantes (UTE-IRS UN) between 2011 and 2015 followed the European Union guidelines for the care and use of laboratory animals in research (approval #01257.03 from the French Ministry of Higher Education and Research (MESR)). The rats were purchased from Charles River Laboratories (L’Arbresle, 69210, France), and the experiments were approved by the ethics committee for animal experiments (CEEA) of the Pays de la Loire Region and registered under the number 2011.38. The non-invasive M5-T2, mildly invasive F4-T2, moderately invasive F5-T1 and deeply invasive M5-T1 tumors were collected after intraperitoneal injection of 3 × 10^6^ cells of the corresponding cell lines (https://technology-offers.inserm-transfert.com/offer/, accessed on 30 January 2023, recorded as RT00418, RT00419, RT00421 and RT00417, respectively) into syngeneic rats.

### 2.2. Proteomic Analyses

For each sample analyzed, four or five 20-µm-thick sections of tumor tissue were scratched with a scalpel and collected in a 1.5-mL Eppendorf^®^ microtube. Next, all the material collected was deparaffinized in three successive xylene washes and then rehydrated in 100%, 95%, 70% and 50% ethanol solutions. The pellets were vacuum-dried, and the dried tissues resuspended in 200 µL of Rapigest SF (Waters, Milford, MA, USA). Dithiothreitol (AppliChem, Darmstadt, Germany) was then added (5 mM final concentration), and the samples were incubated in a thermo shaker at 95 °C for one hour before being sonicated twice (ultrasonic processor 75185, Bioblock Scientific, Illkirch, France). Cystein residues were alkylated by adding 200 mM S-Methyl methanethiosulfonate at 37 °C (10 mM final concentration). Sequencing-grade trypsin was added at a ratio ≥ 2 µg mm^−3^ tissue (at 37 °C overnight). The reaction was stopped with formic acid (9% final concentration, incubation at 37 °C for one hour), and the acid-treated samples were centrifuged at 16,000× *g* for 10 min. After removing the salts from the supernatant, the peptides were collected in a new Eppendorf^®^ microtube using C18 STAGE tips, and their concentration finally determined using the Micro BCA™ Protein Assay Kit (Thermo Fisher Scientific, St Herblain, France). The rat spectral library, SWATH-MS analysis, peptide identification, and relative quantification were performed as previously described [15]. The statistical analysis of the SWATH data set and peak extraction output data matrix from PeakView were imported into MarkerView (v.2, AB Sciex Pte, Ltd., Framingham, MA, USA) for data normalization and relative protein quantification. Proteins with a statistical *p*-value < 0.05, estimated by MarkerView, were considered to be differentially expressed under different conditions.

### 2.3. Histology and Immuno-Histochemical Analyses

The FFPE blocs were cut with a Leica RM2255 microtome (Leica Biosystems, Nussloch, Germany). Areas of interest for both proteomic and histological analyses were selected based on examination of sections of all samples stained with hematoxylin phloxine saffron (HPS), scanned on a Nanozoomer 2.0 HT Hamamatsu. For immuno-histochemistry, tumor sections were stained with anti-ACADL NBP2-92854 polyclonal antibody (Novus Biologicals, Centennial, CO, USA).

### 2.4. Chemicals

Cell line culture medium and fetal bovine serum (FBS) were from Invitrogen Life Technologies (Carlsbad, CA, USA). Cell culture plasticware was from Falcon (Becton Dickinson, Hongkong, China). A BCA Kit from Sigma Chemical Co. (Saint Louis, MO, USA) was used to determine protein contents. Reagents for electrophoresis were bought from Bio-Rad Laboratories. All the other reagents, unless otherwise specified, were purchased from Sigma Chemical Co.

### 2.5. Cells

Ten primary human MM cell lines (5 epithelioid and 5 sarcomatoid), obtained during diagnostic thoracoscopies, were collected from the S. Antonio e Biagio e Cesare Arrigo Hospital Biological Bank of Malignant Mesothelioma (Alessandria, Italy) after obtaining written informed consent. The local Ethics Committees approved the study (#9/11/2011; #126/2016). Primary MM cells were used until passage 10. Table 1 contains clinical and pathological data of the MM patients. Primary MM cells were cultured in HAM’s F12 medium and supplemented with 10% *v*/*v* fetal FBS and 100 U/mL penicillin-100 μg/mL streptomycin.

### 2.6. Immunoblotting

Cells were rinsed with lysis buffer (150 mM NaCl; 1.0% Nonidet P-40; 50 mM Tris-Cl; pH 7.4), supplemented with the protease inhibitor cocktail, sonicated and centrifuged (13,000× *g*, for 10 min at 4 °C). Then, 20 μg of proteins were probed with antibodies ACADL (ab152160, Abcam, Cambridge, UK), GAPDH (sc-47724, Santa Cruz Biotechnology Inc., Dallas, TX, USA) and then with secondary antibodies conjugated with peroxidase (Bio-Rad Laboratories, Hercules, CA, USA). After washing blots with Tris-buffered saline/Tween 0.01% *v*/*v*, blots were developed with enhanced chemiluminescence (Bio-Rad Laboratories) and visualized using a ChemiDoc^TM^ Touch Imaging System device (Bio-Rad Laboratories).

### 2.7. Mitochondria Isolation

Cells were washed twice with PBS, then lysed in 0.8 mL of mitochondria lysis buffer (50 mM TRIS, 100 mM KCl, 5 mM MgCl_2_, 1 mM EDTA and 1.8 mM ATP, pH 7.2) mixed with protease inhibitor cocktail set III (100 µL). PMSF (100 µL) and NaF (25 µL). Cells were scraped and collected in an Eppendorf^®^ tube and then sonicated twice for 10 s at 40% power. Subsequently, samples were centrifuged at 2000 rpm for 1 min at 4 °C. The supernatant was collected into a new series of Eppendorf^®^ tubes and centrifuged again at 13,000 rpm for 5 min at 4 °C. Pellets containing mitochondria were washed with 0.4 mL of mitochondria lysis buffer and centrifuged at 13,000 rpm for 5 min at 4 °C. Subsequently, the supernatant was aspirated and the pellets resuspended in 0.2 mL of mitochondria resuspension buffer (Sucrose 250 mM, K_2_HPO_4_ 15 mM, MgCl_2_ 2 mM and EDTA 0.5 mM, pH 7.2). The resuspended mitochondria were then divided into two parts: one part was used to measure mitochondria protein content using a BCA kit (Sigma, Saint Louis, MO, USA), and the other was divided into 50 µL aliquots and stored at −80 °C until use.

### 2.8. ETC (Electron Transport Chain from Complex I to Complex III)

The electron transport between complexes I and III was measured in mitochondrial extracts obtained previously. In particular, 10 µL of mitochondria samples were put in a 96-well plate, together with 160 µL of buffer A (5 mM KH_2_PO_4_, 5 mM of MgCl_2_, 5% *w*/*v* bovine serum albumin, pH 7.2), 100 µL of buffer B (50 mM KH_2_PO_4_, 5 mM MgCl_2_, 5% *w*/*v* serum bovine albumin, 0.05% saponin, pH 7.5) and freshly added 0.12 mM of cytochrome c-oxidized form and 0.2 mM of NaN_3_. After waiting 5 min to equilibrate the plate at room temperature, 30 µL of NADH (0.15 mM and diluted in buffer B) was added to each well. The reaction then started, and the absorbance was read at 550 nm for 6 min, with 1 read every 15 s. Considering only the linear part of the curve and calculating results in accordance with Lambert-Beer equations, the results obtained were expressed as nmoles of cytochrome C reduced/min/mg mitochondrial protein.

### 2.9. ATP

ATP quantities were measured following the Sigma-Aldrich protocol 213-579-1. First 50 µL of ATP assay mix (lyophilized powder containing luciferase, luciferin, MgSO_4_, DTT, EDTA, BSA and tricine buffer salts, pH 7.8) was added to a vial for 3 min. Then, 50 µL of sample (mitochondria extract obtained as described in the previous steps) was rapidly added and the quantity of light was measured in a black 96-well plate in a microplate reader. The results were expressed as nmols of ATP/mg mitochondrial.

### 2.10. β-Oxidation of Fatty Acid

Assays were performed using the fatty acid complete oxidation kit (ab222944; Abcam, Cambridge, UK) as per the manufacturer’s instructions. Cells were plated at 40,000 cells per well in a 96-well plate with 200 µL of medium per well and left overnight to equilibrate. The cells were washed twice with prewarmed FA-free measurement media, incubated with FA measurement media (150 µM FAO-Conjugate; 0.5 mM L-Carnitine) with extracellular O_2_ consumption reagent (ab197243; Abcam, Cambridge, UK) and then sealed with mineral oil. The fluorescence signal was read in a microplate reader (Ex/Em = 380/650 nM). The results were expressed as pmoles of O_2_/min.

### 2.11. Scratch Assay

Cells were plated at 1 × 10^6^ cells per well in a 6-well plate. After 24 h, scratches using a 20–200 µL pipette tip were made. Cell migration was calculated measuring distance (in µM) between the cells at T0 (immediately after the scratch) and T1 (24 h after the scratch) and dividing it by 24 h. The results were expressed as µM/h.

### 2.12. Real Time PCR (RT-PCR)

Total RNA was extracted using VWR Life Science RiboZol™ RNA Extraction Reagent (VWR Life Science, Radnor, PA, USA) and reverse-transcribed using the iScriptTM cDNA Synthesis Kit (Bio-Rad Laboratories). qRT-PCR was carried out using SYBR Green Supermix (Bio-Rad Laboratories). qPrimerDepot software (http://primerdepot.nci.nih.gov/, accessed on 13 september 2022) was used to obtain the desired PCR primers (Appendix A). Gene Expression Quantitation software (Bio-Rad Laboratories) was used to assess relative gene expression levels.

### 2.13. Statistical Analysis

All data in the text and figures are provided as means ± SEM. The results were analyzed using a one-way ANOVA and Tukey test. *p* < 0.05 was considered significant.

## 3. Results

### 3.1. Mitochondrial Biomarkers Involved in the Acquisition of Invasiveness in Rat Mesotheliomas

To identify a set of mitochondrial proteins involved in the acquisition of tumor invasiveness, we analyzed the proteomes of four experimental models of mesothelioma grown in immunocompetent F344 rats presenting increasing stages of invasiveness. For each tumor type, 1300 proteins were detected, and the comparison of abundance levels for the three invasive tumors ((1) mildly invasive F4-T2, (2) moderately invasive F5-T1 and (3) deeply invasive M5-T1)) versus (4) the noninvasive tumor M5-T2 (Figure 1) produced a list of 433 proteins satisfying the condition *p* < 0.05. The full list of genes encoding these proteins, together with their full names, is given in Appendix A.

In a second step, the subcellular extracellular locations of these proteins were recorded on https://www.proteinatlas.org (accessed on 29 September 2022), and 36 proteins exclusively or mainly located in mitochondria were identified. A list of mitochondrial proteins exhibiting significant abundance changes (increase or decrease in [1 + 2 + 3] vs. 4) is shown in Table 2.

Twenty-six proteins were reported to be exclusively located in mitochondria, including 17 concerned with the most dramatic changes (13 increased and 4 decreased, with *p* < 0.01) and involved in 11 main biological functions. Among the proteins increasing in abundance with invasiveness, two were involved in fatty acid β-oxidation (FAO), encoded by *Acadl* [16] and *Hsd17b10* (Table 1 and Figure 2A) [17], and one in adenine nucleotide metabolism (encoded by *Ak2*) [18]. This list also included two subunits of ATP synthase, with the first being one of the F0 membrane-spanning components (proton channel) (encoded by *Atp5h*) [19] and the second being part of the connector linking the F1 catalytic core to F0 (encoded by *Atp5o*) [20]. Other increased proteins corresponded to two subunits of the cytochrome c oxidase (encoded by *Mtco2*, *Cox5b*, *Cox6c2*) [21], a chaperone regulating cellular stress responses (encoded by *Trap1*), two subunits of the isocitrate dehydrogenase (encoded by *Idh3a* and *Idh3b*) [22] and the malate dehydrogenase (encoded by *Mdh2*) [23], and two mitochondrial scaffolding/chaperone proteins (encoded by *Phb* and *Phb2*) [24]. The last two increased proteins participate in protein translation in mitochondria and contribute to mitochondrial genome stability and biogenesis (encoded by *Tufm* and *Ssbp1*, respectively) [25,26]. Of the four main proteins decreasing in abundance with invasiveness, two represented enzymes of sulfur metabolism (encoded by *Suox* and *Mpst*) [27], one was a peroxide reductase playing a role in protection against oxidative stress (encoded by *Prdx3*) [28] and the other modulated ion channels and receptors (encoded by *S100a10*) [29].

Of the enzymes involved in mitochondrial FAO and detected in proteomic analyses, the long-chain acyl coenzyme A dehydrogenase (encoded by *Acadl*) exhibited the most dramatic changes, with a significant increase being observed for each individual invasive tumor (1 vs. 4, 2 vs. 4, 3 vs. 4) (Figure 2A). Another enzyme in this metabolic pathway, also involved in branched-amino acid catabolism and encoded by *Hsd17b10*, exhibited a similar pattern of increase with invasiveness (Figure 2A). Conversely, for two additional enzymes in this pathway (encoded by *Acads*, and *Hadh*), invasiveness was associated with a decrease, while no significant change was observed for each individual comparison, i.e., 1 vs. 4, 2 vs. 4, and 3 vs. 4 for ACADS (Figure 2A).

The evolution in ACADL and HCD2 levels was associated with a parallel increase in two subunits of ATP synthase (Figure 2B) and three subunits of cytochrome oxidase (Figure 2C), suggesting a link with ATP production and flux within the electron transport chain. A similar increased level of two enzymes in the tricarboxylic cycle, i.e., malate dehydrogenase 2 and isocitrate dehydrogenase, also tended to demonstrate its involvement in the invasiveness process (Figure 2D).

### 3.2. Immuno-Histochemical Study of ACADL Distribution in Rat Tumors

Examination of ACADL expression by IHC in the four tumor models revealed pronounced differences with the level of invasiveness. The non-invasive tumor (M5-T2) was characterized by the absence of staining (Figure 3A). In contrast, the mildly invasive F4-T2 (Figure 3B) and moderately invasive F5-T1 (Figure 3C) tumors exhibited a weak, homogeneous distribution of ACADL expression within the tumor tissues. The most striking feature was the strong staining observed in the deeply invasive M5-T1 tumor (Figure 3D). Moreover, ACADL expression appeared heterogeneous within the tumor, with some areas showing intense staining in external parts of the tumor, as shown on high magnification views (Figure 3E).

### 3.3. Fatty Acid β-Oxidation Supports Cell Invasiveness in Human Primary Mesothelioma Cell Lines

Next, to determine whether our findings on rat mesothelioma tumors could be confirmed in human malignant mesothelioma (MM), we analyzed the differential expression of genes encoding the different mitochondrial proteins of interest listed above (in Section 3.1, Table 1) in five primary sarcomatoid and five primary epithelioid mesothelioma cell lines. Interestingly, the most impressive increase was observed in the expression of *ACADL*. Other highly expressed genes in sarcomatoid mesothelioma cell lines were two ATP synthase subunits (*ATP5H*, *ATPO*), *MTCO2*, *COX5B*, *COX 6C2*, *IDH3A*, *IDH3B* and *TIMM9* (Figure 4A). These findings confirm proteomic data obtained in rat tumors. To evaluate the role of ACADL in the migration/invasiveness of mesothelioma cells, we chose two primary epithelioid mesothelioma cell lines (UP1, UP2) and two primary sarcomatoid mesothelioma cell lines (UP 6, UP 7) derived from patients with the highest and lowest OS, respectively (Table 1), which were therefore indicative of higher or lower invasive properties. In agreement with this finding, primary mesothelioma cells were characterized by low and high migration rates, respectively (Figure 4B,C). Migration of primary mesothelioma cells, evaluated with a scratch assay, was inhibited by addition of etomoxir, a drug that blocks FAO (Figure 4B,C). Primary sarcomatoid mesothelioma cell lines have higher expression of ACADL mRNA (Figure 5A) and protein (Figure 5B), accompanied by higher activity of FAO in comparison with epithelioid mesothelioma cell lines (Figure 6A). Etomoxir did not change *ACADL* expression (Figure 5A,B) but it functionally inhibited FAO in the primary sarcomatoid mesothelioma cell lines (Figure 6). A higher FAO rate (Figure 6A) fuels the electron transport chain, which works faster (Figure 6B) and causes higher ATP production (Figure 6C). In addition to higher FAO, primary sarcomatoid mesothelioma cell lines have more active mitochondrial respiratory complexes and produce more ATP. All these metabolic processes are inhibited by etomoxir (Figure 6). Altogether, these data confirm that FAO supports ATP production through electron transport chain activity, providing energy for cell migration/invasiveness in sarcomatoid mesothelioma tumors.

## 4. Discussion

Cross-species investigations have provided new insights into universal mechanisms in biology, improving, for example, our understanding of oncogenic signatures in breast cancer development in humans and dogs [30]. Applied to proteomic analyses in cancer, common biomarkers of invasiveness have been identified in rat and human mesotheliomas [14]. Genomic analyses have also pointed to markers which are useful for the diagnosis and prognosis of hepatocellular carcinomas in both species [31]. To date, cross-species comparisons of important findings relevant to mitochondria have been very limited, focusing, for example, on detecting heteroplasmy [32]. In this study, we identified several biomarkers of interest that appear to play an important role in metabolic rewiring and invasiveness in both human and rat mesotheliomas.

As biosynthetic hubs, mitochondria consume a variety of different fuels to generate energy in the form of ATP for cancer cells, where fatty acid oxidation plays an important role [33]. Although most cancer researchers initially focused on glycolysis, glutaminolysis and fatty acid synthesis, the relevance of fatty acid oxidation in the metabolic reprogramming of cancer cells was extensively reviewed 10 years ago, and its role in NADPH production was emphasized [34]. Linked to this statement, our results revealed a consistent finding regarding the FAO enzyme *ACADL*, observed both in humans and rats, and associated with the acquisition of invasive properties, i.e., higher expression of *ACADL* was initially found to be positively correlated to prostate cancer progression [35].

Our data also agreed with the work of Yu et al. showing that *ACADL* was overexpressed both in cell lines and clinical specimens, being related to esophageal squamous cell carcinoma progression and poor prognosis [16]. Another close FAO enzyme, which is encoded by *HSD17B10*, is also involved in branched amino acid catabolism and steroid metabolism. Our data are in line with previously published reports emphasizing its upregulation in invasive tumors. For example, Salas et al. showed its predictive value in the response to chemotherapy in osteosarcomas [17]. Its overexpression also accelerated cell growth, enhanced cell respiration and increased cellular resistance to cell death in pheochromocytoma [36]. Finally, and even more interestingly, Condon et al. found that *HSD17B10* was one of the six genes impacting the mTORC1 pathway [37], which is dysregulated and activated in cancer cells to drive survival, neovascularization and invasion [38].

Interestingly, the increased β-oxidation rate, electron flux and ATP production observed in human sarcomatoid mesothelioma cell lines were all consistent with the increased expression of ATP synthase subunits, cytochrome *c* oxidase subunits, abundance changes in these proteins in rat tumors, and with our observations concerning the long-chain acyl coenzyme A dehydrogenase. Fiorillo et al. have highlighted the fact that ATP-high cancer cells are phenotypically the most aggressive, with enhanced stem-like properties, multi-drug resistance potential and an increased capacity for cell migration, invasion and metastasis [9]. Wang et al. also pointed out that high ATP expression was linked to poor prognosis in glioblastoma, clear cell renal cell carcinoma and ovarian, prostate, and breast cancers [39]. Moreover, an additional role of ATP synthase in the formation of the permeability transition pore (PTP) was also recently reported as representing a mechanism controlling tumor cell death [40]. In this process, our findings also tend to confirm the important role of the subunit *d* of ATP synthase (encoded by (*Atp5h*/*ATP5H*), linked to the work by Chang et al., who reported the involvement of the overexpression of this subunit in venous invasion, distant metastasis of colon cancer and, finally, poor survival [41].

Within the enzymes of mitochondrial metabolism involved in cancer progression, besides isocitrate dehydrogenase and malate dehydrogenase, subunits of the cytochrome *c* oxidase (complex IV of the respiratory chain) such as COX5B have also been reported [42]. Our results agreed with previously published literature on the impact of its high expression on tumor invasiveness and poor prognosis in patients with breast cancer [43]. More recently, further insights have confirmed its tremendous role as a growth-promoting gene, both in hepatoma [44] and colorectal cancer [45]. Interestingly, the combined upregulation of COX5B and ATP5H was also reported by Yusenko et al. in renal oncocytomas [19]. Another subunit of the cytochrome *c* oxidase, COX6C, also upregulated in relation to invasiveness in our study, appeared to be differentially expressed in various cancers [46]. Notably, Jang et al. detected it in extracellular vesicles (EV) in the plasma of metastatic melanoma and ovarian and breast cancer patients, suggesting that the classic EV production and mitochondrial pathways are interconnected [47]. In that study, an additional crucial observation was the presence of another inner mitochondrial membrane protein in these EVs [47], encoded by *MTCO2*. These breakthroughs are consistent with both the increased abundance of SODM and the expression of this gene that we found in the most invasive rat tumors as well as in human mesothelioma cell lines. Linked to the tremendous increase in ACADL, the greater abundance and expression of the two subunits of isocitrate dehydrogenase tend to confirm previous observations regarding the central role of the TCA cycle in metabolic reprogramming and tumor invasiveness. Laurenti and Tennant have previously reviewed the impact of its dysregulation in cancers in association with hypoxia and increased intracellular levels of ROS [48]. Moreover, as shown by Zeng et al., the aberrant expression of *IDH3A*, which represented an upstream activator of HIF-1, promoted tumor growth and angiogenesis in various cancer types [22].

In addition to the dramatic changes observed in ACADL associated with tumor invasiveness, we also identified another protein involved in the mitochondrial translation machinery, i.e., *TUFM*. This observation, which is consistent with the higher abundance and expression of proteins involved in mtDNA maintenance, may be relevant to data from several existing reports. For example, Cruz et al. found this protein in a list of five candidate biomarkers of drug-resistant ovarian cancer [49]. Interestingly, the mitochondrial translation pathway is required for increased electron transport chain activity [50], and its inhibition plays a part in sensitizing renal cell carcinoma to chemotherapy [25]. Chatla et al. demonstrated that TUFM was required for increased mitochondrial biosynthesis [51]. Moreover, the authors of that work suggested the existence of a link with another elevated mitochondrial protein found in our study, i.e., encoded by *ALDH7A1*. ALDH7A1 is an enzyme which mechanistically appeared to provide cells with protection against various forms of stress through multiple pathways [52]. It is involved in stem cell pathways [53,54], and the link between its high expression and tumor invasiveness has been clearly established through the works of van den Hoogen et al. [55] and Giacalone et al. [56] in prostate cancer and lung cancer, respectively. Interestingly, in good agreement with our findings, Lee et al. also demonstrated its relationship with lipid catabolism as an energy source in pancreatic cancer cells [57]. ALDH7A1 was first known as antiquitin; the study of its subcellular localization revealed its presence in cytosol in addition to mitochondria [58]. Finally, an intriguing feature of this enzyme, which resonates with the latter observation, was presented in a recent work by Babbi et al., i.e., the central role played by this protein, which is also present in the nucleus, is to interact with 23 other proteins in IntAct and 62 in BioGRID, while *ALDH7A1* represents one of the most frequent genes in KEGG metabolic pathways [59].

## 5. Conclusions

In conclusion, starting from a proteomic approach and following on with ad hoc biological validation, we identified significant differences between non-invasive and invasive mesotheliomas, developed in both rats and patient-derived cells, in terms of the expression of mitochondrial proteins. This suggests that mitochondrial activity plays an important role in cancer. In particular, ACADL and subunits of ATP synthase are highly expressed in invasive rat mesotheliomas, as well as in more aggressive human sarcomatoid mesothelioma cells, which have more active FAO, electron chain transport and ATP synthesis, supporting their growth and invasiveness. Evaluating mitochondrial proteins in MM specimens might help to identify tumors with higher invasiveness and new potential targets that could be explored to improve the treatment of this disease.

## Figures and Tables

**Figure 1 cancers-15-03044-f001:**
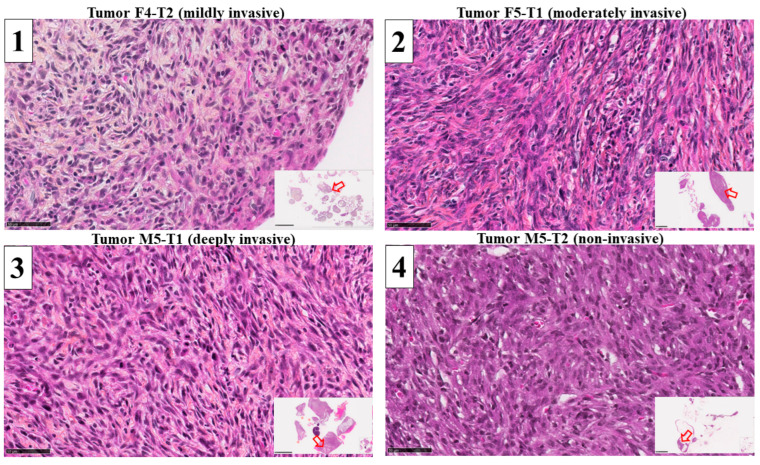
Histological features of the four experimental rat mesothelioma tumor models. HPS staining, ×400 (the scale bar represents 50 µm). Inserts (bottom right corner) represent general views (the scale bars represent 5 mm (left column) or 2.5 mm (right column)), with the open red arrows showing the location of the magnified areas.

**Figure 2 cancers-15-03044-f002:**
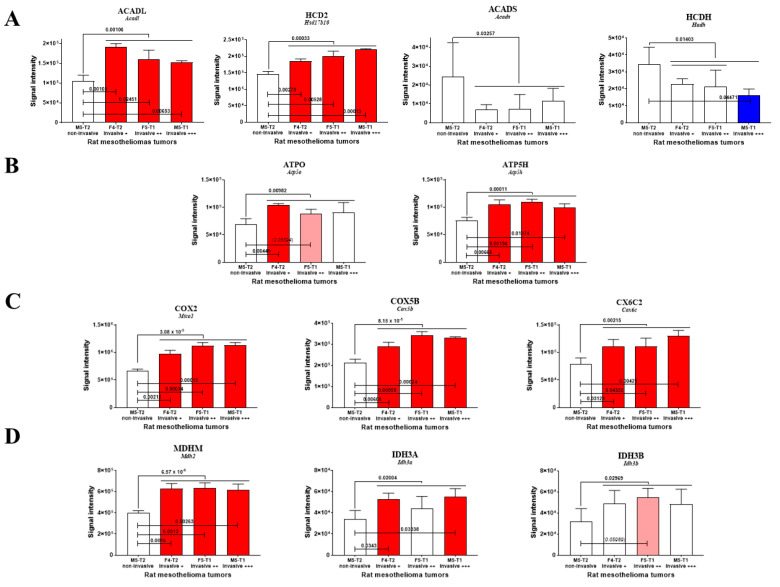
Abundance changes with invasiveness, main mitochondrial proteins. (**A**) FAO enzymes. (**B**) ATP synthase subunits. (**C**) Cytochrome oxidase subunits. (**D**) TCA enzymes. Red bars represent increase and blue bars decrease, with light colors used for tendencies. Protein codes (for *rattus norvegicus*) are put in upper case and bold, and gene names in italics.

**Figure 3 cancers-15-03044-f003:**
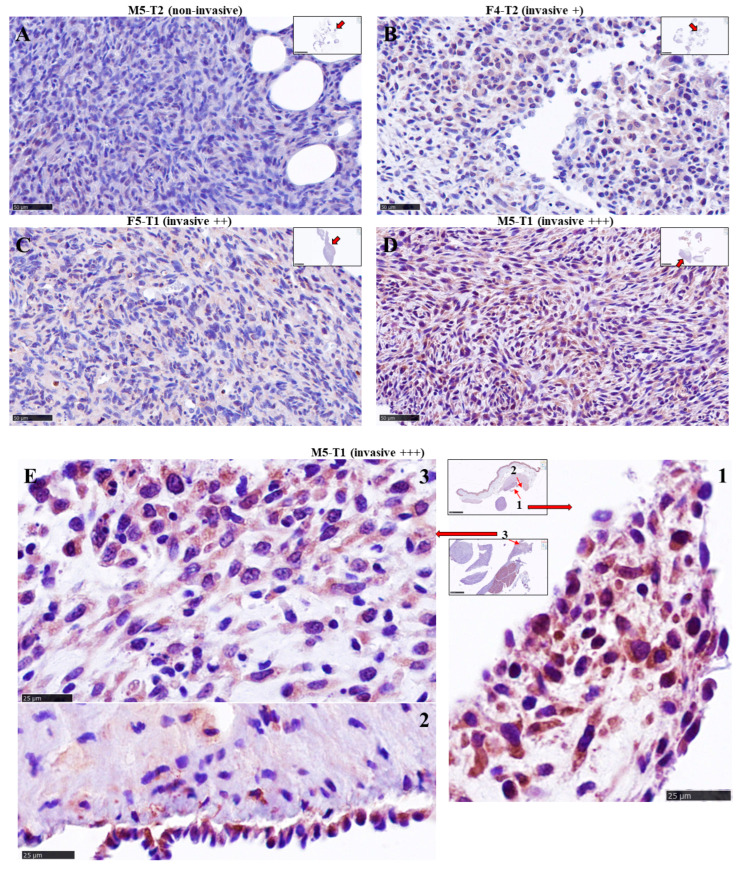
Distribution of *ACADL* expression in rat mesothelioma tumors. (**A**–**D**) Comparison of overall IHC staining with increasing invasiveness, ×400 (the scale bars represent 50 µm). (**E**) Magnifications of areas of intense staining in the most aggressive, M5-T1 tumor (the scale bars represent 25 µm).

**Figure 4 cancers-15-03044-f004:**
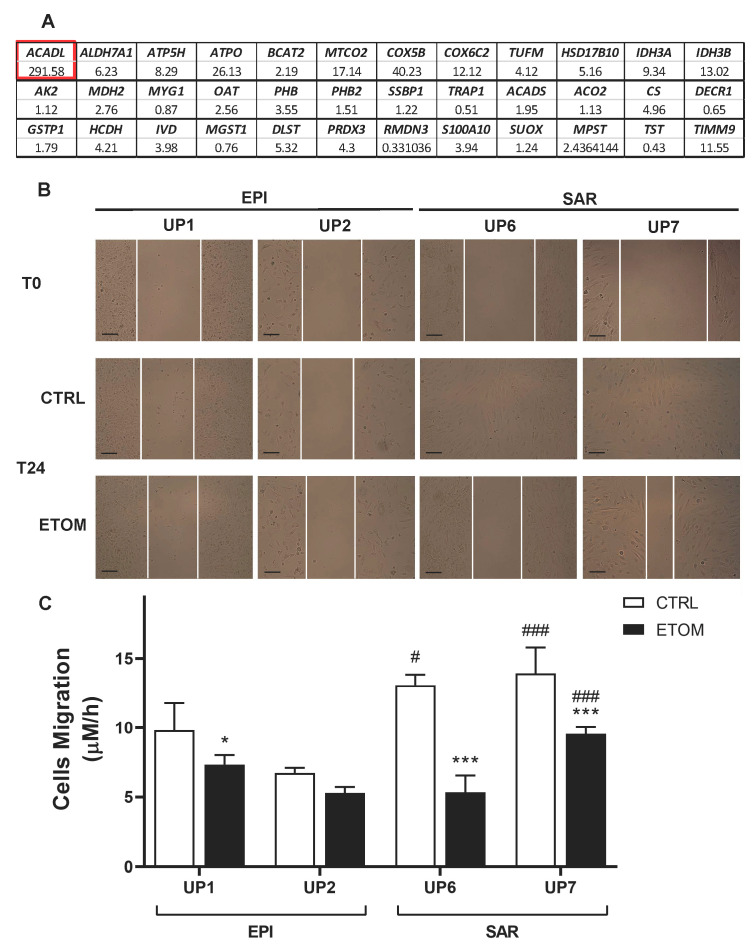
Different expressions of mitochondrial genes between epithelioid and sarcomatoid MM cells. (**A**) Mitochondrial gene expression in 10 primary MM cell lines (Table 1) derived from two different histopathological subtypes, i.e., epithelioid (EPI, *n* = 5) and sarcomatoid (SAR, *n* = 5), was analyzed with real time PCR. Data are expressed as relative mean fold increase SAR vs. EPI MM cells. (**B**,**C**) MPM epithelioid (EPI UP1 and EPI UP2), and sarcomatoid (SAR UP6 and SAR UP7) cells were grown to confluence, then scratched and incubated for 24 h in fresh medium (CTRL) or medium with 10 µM of etomoxir (ETOM). (**B**) Representative bright-field images immediately after the scratch and after 24 h. (**C**) Cell migration. Data are presented as means ± SEM (*n* = 3). * *p* < 0.05, *** *p* < 0.001: ETOM treated cells vs. CTRL cells; # *p* < 0.05, ### *p* < 0.001: SAR cells vs. EPI cells. Scale bar is 100 µm.

**Figure 5 cancers-15-03044-f005:**
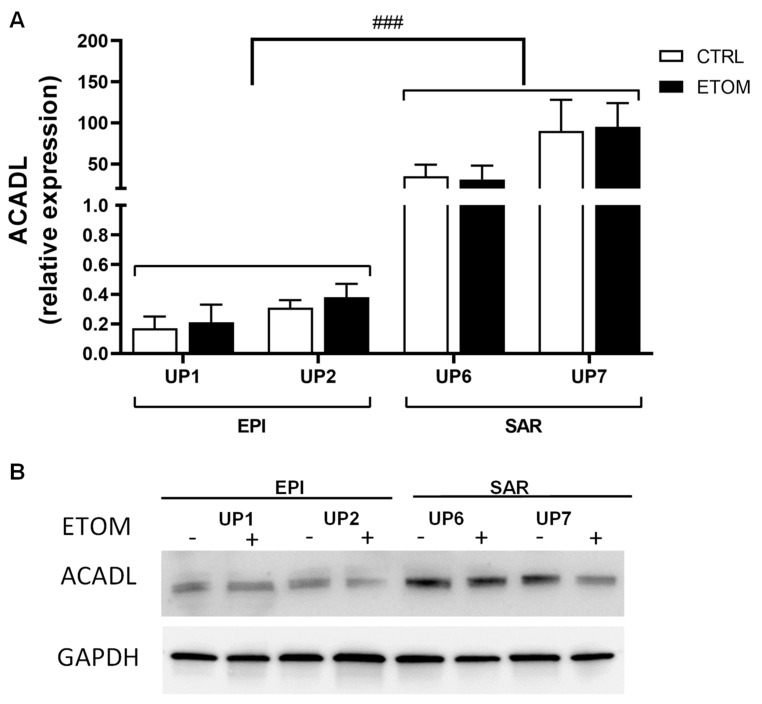
Sarcomatoid MPM cells have higher expression of *ACADL* compared with epithelioid MM cells. Primary MM cells derived from two different histopathological subtypes, i.e., epithelioid (EPI UP1 and UP2) and sarcomatoid (SAR UP6 and UP7), were incubated in fresh medium (CTRL), or in medium with 10 µm of etomoxir (ETOM) for 24 h then used for measurements. (**A**) *ACADL* mRNA levels were measured with RT-PCR, in triplicate. Data are presented as means ± SEM (*n* = 3). ### *p* < 0.001: SAR cells vs. EPI cells. (**B**) *ACADL* protein was measured with immunoblotting in primary MM cell lines. GAPDH was used as a loading control. The figure is representative of one out of three experiments with similar results. The uncropped blots and molecular weight markers are shown in Appendix A.

**Figure 6 cancers-15-03044-f006:**
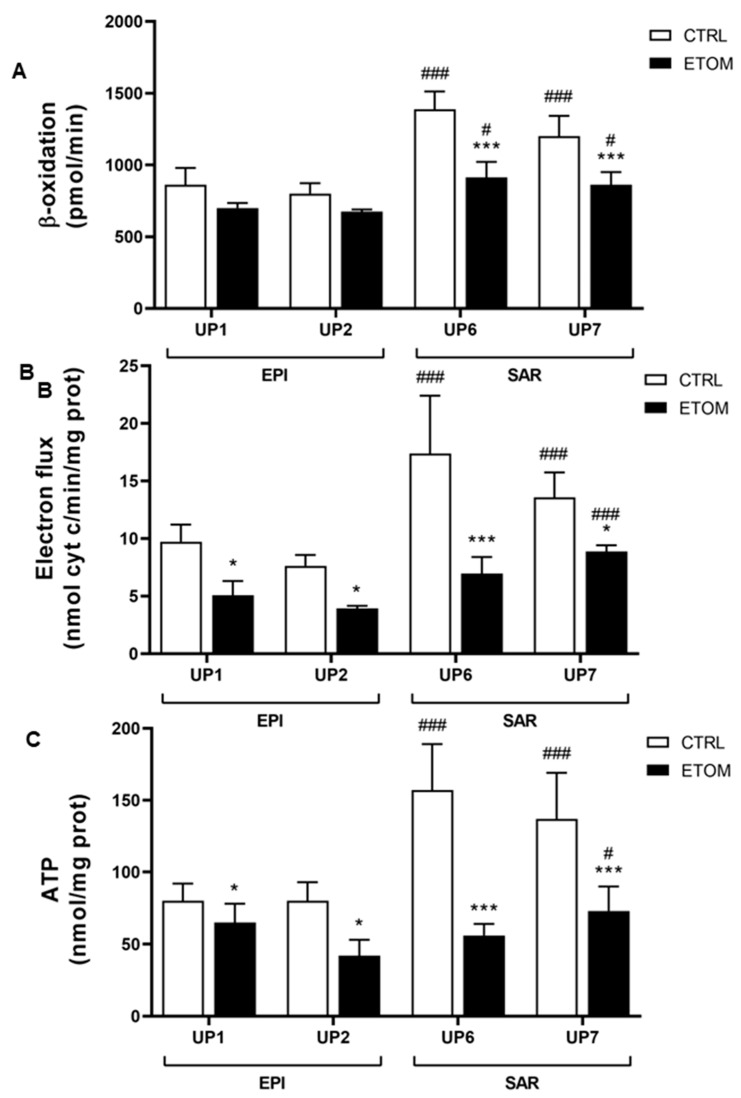
Sarcomatoid MM cells have more active mitochondrial metabolism compared with epithelioid MM cells. Primary MM cells derived from two different histopathological subtypes, i.e., epithelioid (EPI UP1 and UP2) and sarcomatoid (SAR UP6 and SAR UP7), were grown in fresh medium (CTRL) or in medium with 10 µM of etomoxir for 24 h and then used for the following analysis. (**A**) Fatty acid β-oxidation was measured with fluorimetric assay in triplicate. Data are presented as means ± SEM (*n* = 3). *** *p* < 0.001: ETOM treated cells vs. CTRL cells; # *p* < 0.05, ### *p* < 0.001: SAR cells vs. EPI cells. (**B**) The electron flux between Complex I and III was measured spectrophotometrically in triplicate. Data are expressed as means ± SEM (*n* = 3). * *p* < 0.05, *** *p* < 0.001: ETOM treated cells vs. CTRL cells, ### *p* < 0.001: SAR cells vs. EPI cells. (**C**) ATP release was measured with a chemiluminescence-based assay in duplicate. Data are expressed as means ± SEM (*n* = 3). * *p* < 0.05, *** *p* < 0.001: ETOM treated cells vs. CTRL cells; # *p* < 0.05, ### *p* < 0.001: SAR cells vs. EPI cells.

**Table 1 cancers-15-03044-t001:** Origin and characteristics of human mesothelioma cell lines.

UNP (Number)	Histotype	Gender	Age (Years)	Asbestos Exposure	First Line of Treatment	Second Line of Treatment	TTP (Months)	OS (Months)
1	Epithelioid	M	74	Unknown	Carbo + Pem	No	7	11
2	Epithelioid	F	58	Yes	Carbo + Pem	Pem	6	13
3	Epithelioid	M	76	Unknown	CisPt + Pem	No	3	8
4	Epithelioid	M	68	Yes	Carbo + Pem	Pem	4	9
5	Epithelioid	F	84	Yes	CisPt + Pem	No	7	8
6	Sarcomatoid	M	80	Yes	Carbo + Pem	Trabectedin	3	5
7	Sarcomatoid	F	78	Unknown	Pem	No	4	6
8	Sarcomatoid	M	69	Yes	Carbo + Pem	Trabectedin	7	10
9	Sarcomatoid	F	74	Unknown	Carbo + Pem	No	5	7
10	Sarcomatoid	M	78	Yes	Carbo + Pem	Trabectedin	4	9

UNP: unknown patient; M: male; F: female; Carbo: carboplatin; Pem: pemetrexed; CisPt: cisplatin; TTP: time to progression; OS: overall survival.

**Table 2 cancers-15-03044-t002:** Mitochondrial proteins exhibiting significant abundance changes (*p* < 0.05) in the three invasive rat malignant mesothelioma tumors relative to the non-invasive tumor. # According to www.uniprot.org for Rattus norvegicus. * Protein location not restricted to mitochondria. ↑ Increased abundance, ↓ decreased abundance.

Code ^#^	Gene ^#^	Full Name ^#^	[1 + 2 + 3] vs. 4
ACADL	*Acadl*	Long-chain specific acyl-CoA dehydrogenase, mitochondrial	↑
AL7A1 *	*Aldh7a1*	Alpha-aminoadipic semialdehyde dehydrogenase	↑
ATP5H	*Atp5h*	ATP synthase subunit d, mitochondrial	↑
ATPO	*Atp5o*	ATP synthase subunit O, mitochondrial	↑
BCAT2 *	*Bcat2*	Branched-chain-amino-acid aminotransferase, mitochondrial	↑
COX2	*Mtco2*	Cytochrome c oxidase subunit 2	↑
COX5B	*Cox5b*	Cytochrome c oxidase subunit 5B, mitochondrial	↑
CX6C2	*Cox6c2*	Cytochrome c oxidase subunit 6C-2	↑
EFTU	*Tufm*	Elongation factor Tu, mitochondrial	↑
HCD2	*Hsd17b10*	3-hydroxyacyl-CoA dehydrogenase type-2	↑
IDH3A	*Idh3a*	Isocitrate dehydrogenase [NAD] subunit alpha, mitochondrial	↑
IDH3B	*Idh3b*	Isocitrate dehydrogenase [NAD] subunit beta, mitochondrial	↑
KAD2	*Ak2*	Adenylate kinase 2, mitochondrial	↑
MDHM	*Mdh2*	Malate dehydrogenase, mitochondrial	↑
MYG1 *	*Myg1*	UPF0160 protein MYG1, mitochondrial	↑
OAT *	*Oat*	Ornithine aminotransferase, mitochondrial	↑
PHB	*Phb*	Prohibitin	↑
PHB2	*Phb2*	Prohibitin-2	↑
SSBP	*Ssbp1*	Single-stranded DNA-binding protein, mitochondrial	↑
TRAP1	*Trap1*	Heat shock protein 75 kDa, mitochondrial	↑
ACADS	*Acads*	Short-chain specific acyl-CoA dehydrogenase, mitochondrial	↓
ACON	*Aco2*	Aconitate hydratase, mitochondrial	↓
CISY *	*Cs*	Citrate synthase, mitochondrial	↓
DECR *	*Decr1*	2, 4 dienoyl-CoA reductase, mitochondrial	↓
GSTP1 *	*Gstp1*	Glutathione S-transferase P	↓
HCDH	*Hadh*	Hydroxyacyl-CoA dehydrogenase, mitochondrial	↓
IVD *	*Ivd*	Isovaleryl-CoA dehydrogenase, mitochondrial	↓
MGST1 *	*Mgst1*	Microsomal glutathione S-transferase 1	↓
ODO2	*Dlst*	Dihydrolipoyllysine-residue succinyltransferase component of 2-oxoglutarate dehydrogenase complex, mitochondrial	↓
PRDX3	*Prdx3*	Thioredoxin-dependent peroxide reductase, mitochondrial	↓
RMD3 *	*Rmdn3*	Regulator of microtubule dynamics protein 3	↓
S10AA	*S100a10*	Protein S100-A10	↓
SUOX	*Suox*	Sulfite oxidase, mitochondrial	↓
THTM	*Mpst*	3-mercaptopyruvate sulfurtransferase	↓
THTR	*Tst*	Thiosulfate sulfurtransferase	↓
TIM9	*Timm9*	Mitochondrial import inner membrane translocase subunit Tim9	↓

## Data Availability

The data can be shared up on request.

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
