# Peer review of "Long-Chain Acyl Coenzyme A Dehydrogenase, a Key Player in Metabolic Rewiring/Invasiveness in Experimental Tumors and Human Mesothelioma Cell Lines"

_cancers, 2023, doi:10.3390/cancers15113044_

Round 1

Reviewer 1 Report

In this manuscript the authors studied the mitochondrial proteins as potential malignant mesothelioma oncomarkers both in experimental rat tumor models and tumor cells from MM patients. The subject of the reviewed manuscript seems to be relevant and topical, the experimental design is elegant, the results are not in doubt, the conclusions are justified. However, important concerns listed below should be addressed to the authors before publication.

Unfortunately, the file uploaded as "Original files of blots/gels" contains the same information as Fig4., so none original images were provided.

The Supplementary Table S3 is identical to Table 1 in the main text of the manuscript.

The mass spec data were not deposited to ProteomeExchange or any other public resources, that does not allow to fully evaluate the presented data.

References: do not meet the criteria established in the journal's “Instructions for Authors” (DOI links can't be opened from outside the authors' university).

And finally, the absence of an abbreviations list also complicates the perception of the material.

Reviewer 2 Report

Cancers #2421707

Reviewer comments

In the work entitled “Long-chain acyl coenzyme A dehydrogenase, a key player in 2 metabolic rewiring/invasiveness in experimental tumors and 3 human mesothelioma cell lines”, the authors used a proteomic scouting approach to identify potential Mitochondrial genes involved in invasive behavior in FFPE tissues obtained from four experimental Mesothelioma animal models (F344 rats injected with Epithelial vs Sarcomatoid Mesothelioma cells at increasing level of invasive behavior against a normal (non-invasive) tissue. This strategy led to the identification of 433 proteins with proper statistical stringency out of which the authors selected the first 20 most expressed and the last 16 least expressed. Long-chain acyl coenzyme A dehydrogenase (ACADL) resulted as the highest expressed. The authors further tested the trend observed in rat mesothelioma models in ten primary mesothelioma cell lines with similar histotypes distribution using RT-PCR and IHC. This approach confirmed the previous trend and further strengthened the role of  ACADL with a fold increase of 7.25 to 133 times compared to other Mitochondrial genes. The authors also confirmed such trend to be marked in the Sarcomatoid (less differentiated) histotype using western blotting and IHC. The authors also complement this main findings with biological testing (scratch assay) using a Fatty Acid Oxidation inhibitor (etomoxir) to further support the relevance of ACADL and the observed invasive behavior in the primary cell models and confirming the higher impact of such functional link in Sarcomatoid Mesothelioma cells.

Overall, the study increase prior knowledge on the role of mitochondrial functions and invasive behavior in cancer. The controls used throughout the study are sufficient. It would have been optimal to extend the number of biological assays (eg invasion by Boyd chambers) or correlation with additional markers of invasiveness or signal pathways involvement in the same models but this does not affects the main message of the work nor the eligibility for publication on the journal considering the bona fide ratio between its current impact factor and the level of originality of the finding. Therefore, this reviewer support the authors claim for publication upon mandatory minor revision for the items reported below.

Minor revision required for the following points:

Add proper columns referencing (eg in parenthesis) in Table 1 [UNP (number or “#”); Age (years or “yrs”); First line (of treatment or “Rx”); TTP and OS (months or “mo”)]. In case of short acronym use, add the full term meaning in figure legend below the table.

Refs-> Need to replace the link for doi used for references (ISERM-linked, not open access) to the corresponding open access link (publisher site or PubMed-linked).
